# From Text to Forecasts: Bridging Modality Gap with Temporal Evolution Semantic Space

**Lehui Li**[1][*]  **Yuyao Wang**[2][*]  **Jisheng Yan**[1][*]  **Wei Zhang**[1]  **Jinliang Deng**[3][†]  **Haoliang Sun**[1]  **Zhongyi Han**[1]
**Yongshun Gong**[1][†]

## Abstract

Incorporating textual information into time-series forecasting holds promise for addressing event-driven non-stationarity; however, a fundamental modality gap hinders effective fusion: textual descriptions express temporal impacts implicitly and qualitatively, whereas forecasting models rely on explicit and quantitative signals. Through controlled semi-synthetic experiments, we show that existing methods over-attend to redundant tokens and struggle to reliably translate textual semantics into usable numerical cues. To bridge this gap, we propose TESS, which introduces a Temporal Evolution Semantic Space as an intermediate bottleneck between modalities. This space consists of interpretable, numerically grounded temporal primitives—distribution shift, volatility, shape, and lag—extracted from text by an LLM via structured prompting and filtered through confidence-aware gating. Experiments on four real-world datasets demonstrate up to a 29% reduction in forecasting error compared to state-of-the-art unimodal and multimodal baselines. Code is available at: https://github.com/olivia3395/TESS.

## 1. Introduction

Time-series forecasting plays a critical role in diverse domains such as transportation, energy, and finance (Zhou et al., 2021; Shao et al., 2024; Wang et al., 2025a). In recent years, unimodal forecasting models that rely solely on

---
[*]Equal contribution . [†]Co-corresponding authors. [1]School of Software, Shandong University, China [2]Department of Mathematics and Statistics, Boston University, Boston, MA, USA [3]State Key Laboratory of Complex & Critical Software Environment, Beihang University. Correspondence to: Jinliang Deng <jinliangdeng9588@gmail.com>, Yongshun Gong <ysgong@sdu.edu.cn>.

*Proceedings of the 43rd International Conference on Machine Learning*, Seoul, South Korea. PMLR 306, 2026. Copyright 2026 by the author(s).

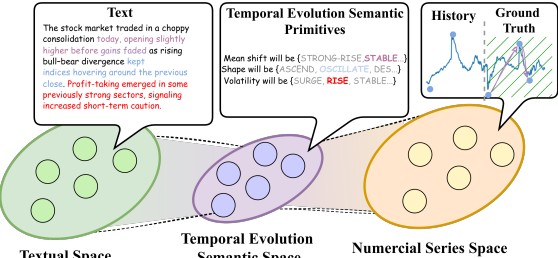

*Figure 1.* Illustration of the cross-modal transformation via the Temporal Evolution Semantic Space. Verbose textual narratives (left) are distilled into structured temporal primitives (middle), which then guide numerical forecasting (right).

historical numerical observations have achieved substantial progress by modeling temporal dependencies (Wu et al., 2021). However, real-world time series often exhibit significant non-stationarity (Du et al., 2021; Kim et al., 2025), where the underlying data-generating mechanism evolves over time. In particular, exogenous events such as accidents, extreme weather, or public sentiment shocks can trigger regime shifts (Osogami, 2021; Yang et al., 2025; Qiu et al., 2025a), causing statistical properties—including trends, volatility, mean, and variance—to change abruptly within short time windows. Under such conditions, models struggle to extract reliable predictive signals from historical numerical observations alone, leading to severe performance degradation (Zeng et al., 2023).

To mitigate event-driven non-stationarity, recent work has begun to incorporate textual data (e.g. news articles, social media posts) (Sawhney et al., 2020; Qiu et al., 2026) as an exogenous information source. The typical approach leverages pretrained language models to encode text into semantic embeddings (Niu et al., 2023), which are then fused with numerical time-series features via concatenation, gating, or cross-modal attention (Sawhney et al., 2020; Koval et al., 2025). These multimodal methods have demonstrated notable gains on benchmarks with significant non-stationarity.

Despite recent progress, the substantial *modality gap* (Liang et al., 2022) between time-series observations and textual information remains largely overlooked (Koval et al., 2025), fundamentally limiting multimodal forecasting. Time-series

observations are chronologically ordered and quantitative, offering precise measurements of temporal dynamics, yet they lack explicit semantic abstraction. In contrast, textual information is semantically rich but unstructured and qualitative, with the impacts of events on temporal dynamics often implicit, diffuse, and weakly grounded in time. As a result, the predictive relevance of textual information is sparsely distributed across tokens and rarely aligned with the compact numerical structure required by time-series models. Through controlled semi-synthetic experiments, we analyze time-series predictors that directly fuse numerical observations with raw textual embeddings. We find that: (1) predictors fail to focus on prediction-relevant tokens, instead being distracted by noisy and redundant textual content; (2) even after removing redundant information, models still struggle to correctly decode temporal evolution signals, indicating a fundamental representational mismatch between the two modalities.

To bridge this modality gap, as illustrated in Figure 1, we propose to construct an intermediate representation between the textual space and the numerical series space, termed the *Temporal Evolution Semantic Space*. This space serves as an information bottleneck that explicitly filters, structures, and organizes textual information relevant to temporal evolution. Inspired by expert-driven paradigms in temporal pattern analysis, we predefine a set of temporal evolution primitives corresponding to critical attributes governing temporal dynamics, such as distribution shift, volatility shift, evolution shape, and lag. Crucially, these primitives are expressed as verbal specifications, enabling large language models (LLMs) to emulate human-like judgments when interpreting textual descriptions of events. We decompose the fusion process into two stages: ①  **Text → Semantic Space**: We prompt an LLM to reason over the input text and explicitly label latent temporal evolution signals using the predefined primitives. Since such labels may be unreliable due to information insufficiency or limitations of the LLM, we introduce a learnable gating network to estimate and weight the reliability of the generated semantic labels. ②  **Semantic Space → Numerical Space**: The inferred temporal evolution primitives are then injected as exogenous conditions into the time-series model, allowing it to leverage its numerical modeling and pattern recognition capabilities to ground semantic signals in observed temporal dynamics. Our **contributions** are summarized as follows:

- Through semi-synthetic experiments, we identify two bottlenecks in text-time-series fusion: attention is distracted by redundant context, and qualitative expressions resist decoding into predictive gains.

- We propose TESS, an intermediate space that distills text into temporal primitives via LLM extraction with confidence-aware gating, guiding forecasters as exoge-

nous conditions.

- Experiments on four datasets show up to 29% MSE reduction; analyses validate primitive complementarity and gating effectiveness across non-stationary patterns.

## 2. Related Work

**Non-Stationary Time-Series Forecasting.** Non-stationary time-series forecasting addresses sequences whose statistical properties evolve over time, representing one of the most challenging scenarios in temporal prediction. Existing approaches enhance stationarity through preprocessing techniques such as decomposition, detrending, differencing, and instance-level normalization (e.g., RevIN) (Wu et al., 2021; Kim et al., 2022; Zhou et al., 2022; Deng et al., 2024b; Qiu et al., 2025b), or introduce explicit non-stationarity modeling at the architectural level, including adaptive drift correction, time-varying attention, and patch-based structures (Liu et al., 2023; Shukla & Marlin, 2021; Nie et al., 2023; Deng et al., 2024a; Qiu et al., 2025c). While these methods improve stability, they remain fundamentally reactive: historical observations alone cannot anticipate exogenous shocks that trigger abrupt distribution shifts, volatility spikes, or structural breaks.

**Multimodal Time-Series Forecasting.** Multimodal time-series forecasting jointly models external text (e.g., news, announcements) with numerical sequences to enhance awareness of event-driven distribution shifts. Early approaches relied on shallow semantic features such as sentiment scores or keyword counts as auxiliary predictors (Tsai & Wang, 2014; Schumaker & Chen, 2009), but such representations lack the granularity to characterize fine-grained temporal dynamics. With pretrained language models, recent work directly fuses raw text via cross-attention or concatenation, yet the fundamental modality gap between qualitative narratives and quantitative signals remains underexplored (Wang et al., 2024b; Koval et al., 2025; Deng et al., 2022). In this work, we systematically investigate this gap and propose an intermediate semantic space that distills text into numerically grounded temporal primitives, enabling stable and effective cross-modal fusion.

## 3. Preliminaries

### 3.1. Problem Formulation

We consider the multimodal time-series forecasting task. Let $\mathbf{x}_t \in \mathbb{R}^d$ denote the numerical observation at time step $t$, and let $s_t$ represent the temporally aligned textual information (e.g., news articles, announcements). Given a historical window of length $L$, the numerical input is defined as

$$\mathbf{X}_{\text{time}} = [\mathbf{x}_{t-L+1}, \ldots, \mathbf{x}_t] \in \mathbb{R}^{L \times d},$$

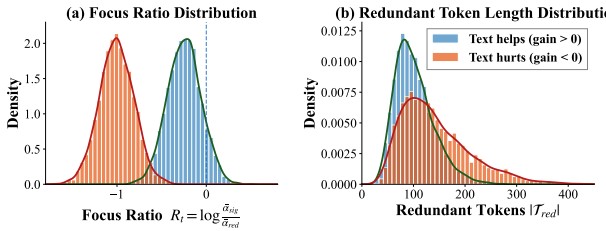

*Figure 2.* Analysis of attention misalignment. **Left**: Distribution of focus ratio $R_t$ on test samples. **Right**: Relationship between redundant token count and predictive performance.

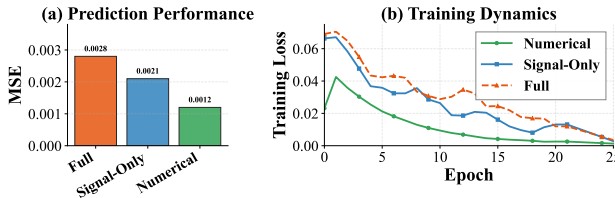

*Figure 3.* Comparison of three input variants. **Left**: Prediction performance (MSE) across Full, Signal-Only, and Numerical inputs. **Right**: Training loss curves showing convergence dynamics.

and the textual input is $\mathbf{X}_{\text{text}} = s_t$. The objective is to predict the numerical sequence over the next $H$ steps:

$$\hat{\mathbf{Y}}_t = [\hat{\mathbf{x}}_{t+1}, \ldots, \hat{\mathbf{x}}_{t+H}] = f_\theta(\mathbf{X}_{\text{time}}, \mathbf{X}_{\text{text}}),$$

where $f_\theta$ denotes a multimodal forecasting model. The prevalent fusion paradigm employs modality-specific encoders to map each modality into latent representations $\mathbf{H}_{\text{time}} = \phi_{\text{time}}(\mathbf{X}_{\text{time}}) \in \mathbb{R}^{N \times d_1}$ and $\mathbf{H}_{\text{text}} = \phi_{\text{text}}(\mathbf{X}_{\text{text}}) \in \mathbb{R}^{M \times d_2}$, where $M$ is the number of text tokens, which are integrated via cross-modal attention or other fusion mechanisms.

### 3.2. Diagnosing the Modality Gap

**Semi-Synthetic Benchmark Construction**    To investigate how the modality gap affects fusion, we construct a semi-synthetic benchmark where text is guaranteed to contain predictive signals, enabling controlled evaluation of information utilization. Specifically, we select FNSPID (Dong et al., 2024), a financial dataset exhibiting pronounced non-stationarity, and extract real numerical sequences as the temporal modality input $\mathbf{X}_{\text{time}}$. For each sample, we further extract a set of statistical features characterizing the evolution patterns from its future window, and leverage GPT-5.2 to map these features into natural language $\mathbf{X}_{\text{text}}$ temporally aligned with the sample.

The traceable generation process enables automatic token-level annotation: tokens encoding future statistical features form the signal set $\mathcal{T}_{\text{sig}}$, while context-only tokens form the redundant set $\mathcal{T}_{\text{red}}$. Since prediction-relevant information is typically sparsely distributed within natural language, we generally have $|\mathcal{T}_{\text{sig}}| \ll |\mathcal{T}_{\text{red}}|$. This design preserves the naturalistic properties of text while introducing controlled interference, enabling us to quantitatively assess in a unified and interpretable setting: when text already contains effective information, whether existing fusion mechanisms can reliably localize $\mathcal{T}_{\text{sig}}$, suppress interference from $\mathcal{T}_{\text{red}}$, and genuinely translate critical semantics into predictive gains.

**Analysis of Attention Distraction**    We first examine whether the fusion mechanism can correctly identify predictive signals within text. To quantify the rationality of attention allocation, we define the focus ratio:

$$R_t = \log \frac{\bar{\alpha}_{\text{sig}}}{\bar{\alpha}_{\text{red}}}, \tag{1}$$

$$\bar{\alpha}_{\text{sig}} = \frac{1}{|\mathcal{T}_{\text{sig}}|} \sum_{i \in \mathcal{T}_{\text{sig}}} \alpha_{t,i}, \ \ \bar{\alpha}_{\text{red}} = \frac{1}{|\mathcal{T}_{\text{red}}|} \sum_{j \in \mathcal{T}_{\text{red}}} \alpha_{t,j},$$

Here $\alpha_{t,i}$ denotes the cross-attention weight on token $i$. If the model correctly localizes predictive signals, we expect $R_t > 0$; conversely, $R_t < 0$ indicates that attention is dominated by redundant information. Figure 2 (**Left**) presents the distribution of $R_t$ on test samples: even among samples where text yields positive predictive gains, the vast majority exhibit $R_t < 0$, indicating that the model systematically over-attends to redundant tokens rather than predictive signals. Figure 2 (**Right**) corroborates this finding: samples with negative gains exhibit significantly more redundant tokens, confirming that redundancy hampers textual information extraction.

**Analysis of Representational Mismatch**    The preceding analysis demonstrates that redundant tokens interfere with signal extraction. A natural question arises: *if redundancy is entirely removed, can the model effectively utilize predictive information in text?* To address this question, we construct three input variants: (1) FULL, containing the complete text (both signal and redundant tokens); (2) SIGNAL-ONLY, retaining only signal tokens $\mathcal{T}_{\text{sig}}$ with all redundancy removed; and (3) NUMERICAL, directly using statistical feature vectors as exogenous inputs. Figure 3 (**Left**) presents the prediction performance comparison across the three variants. SIGNAL-ONLY achieves lower MSE than FULL, confirming the interference effect of redundant information on prediction; however, its performance remains significantly inferior to NUMERICAL, indicating that even when redundancy is completely removed, textual signals still cannot be effectively translated into predictive gains. Figure 3 (**Right**) further corroborates this finding through training curves: NUMERICAL converges rapidly with smooth trajectories, SIGNAL-ONLY exhibits pronounced convergence lag and loss oscillation, while FULL displays the most unstable optimization dynamics. These observations collectively reveal the fundamental bottleneck of cross-modal transformation—

natural language tends to characterize temporal evolution in implicit, qualitative terms (e.g., "significant rise" rather than "+15.3%"), and such semantics are difficult for existing fusion mechanisms to decode into quantitative signals usable for numerical prediction.

## 4. Methodology

To bridge the modality gap, as illustrated in Figure 4, we propose TESS, a two-stage framework centered on the *Temporal Evolution Semantic Space*. This intermediate representation distills natural language into quantifiable temporal primitives, enabling stable cross-modal fusion. We first introduce the design of the temporal evolution semantic space (Section 4.1), then describe the text-to-semantic primitive projection with confidence-aware gating (Section 4.2), and finally present semantic-primitive-conditioned forecasting (Section 4.3).

### 4.1. Temporal Evolution Semantic Space

To overcome the limitations of directly mapping text to numerical sequences, we introduce an intermediate representation—the *Temporal Evolution Semantic Space*—that mediates between textual and numerical modalities. The design rationale derives from established practices in domain-specific forecasting: rather than processing descriptive details verbatim, practitioners extract implications for temporal statistical properties. Guided by this insight, we distill three categories of statistically critical features from the classical forecasting literature and formalize them as four *Temporal Semantic Primitives* (TSPs). The distribution primitives characterize shifts in statistical properties, the shape primitive characterizes *what* aspects of the time series evolve, and the lag-and-decay primitive captures *when* and *for how long* these impacts manifest.

**(1) Distribution Shift Primitives.** Distribution shift—temporal changes in statistical properties—is identified by Kim et al. (2022) as a principal source of forecasting degradation in non-stationary time series. We characterize such shifts using two complementary primitives capturing changes in mean level and volatility. Let $\mathbf{X}_t \in \mathbb{R}^L$ and $\mathbf{Y}_t \in \mathbb{R}^H$ denote the observation and forecast windows. The distribution shift is quantified by the standardized difference $\Delta_\mu = (\bar{Y}_t - \bar{X}_t)/\sigma(X_t)$, where $\bar{X}_t = \frac{1}{L}\sum_i x_i$ and $\bar{Y}_t = \frac{1}{H}\sum_j y_j$.

Volatility shift is measured by $r_\sigma = \log[(\sigma_Y + \epsilon)/(\sigma_X + \epsilon)]$, where $\sigma_X = \text{std}(\Delta\mathbf{X}_t)$, $\sigma_Y = \text{std}(\Delta\mathbf{Y}_t)$, and $\Delta$ denotes the first-order differencing operator. Both shift measures are discretized using adaptive thresholds $\tau_1 < \tau_2$ defined by training-set quantiles.

| | Category | Condition | Semantics |
|---|---|---|---|
| ⇑ | STRONG-RISE | $z \in \{\Delta_\mu, r_\sigma\}, z > \tau_2$ | Significant increase |
| ↑ | MILD-RISE | $z \in \{\Delta_\mu, r_\sigma\}, \tau_1 < z \leq \tau_2$ | Moderate increase |
| — | STABLE | $z \in \{\Delta_\mu, r_\sigma\}, -\tau_1 \leq z \leq \tau_1$ | Level unchanged |
| ↓ | MILD-DROP | $z \in \{\Delta_\mu, r_\sigma\}, -\tau_2 \leq z < -\tau_1$ | Moderate decrease |
| ⇓ | STRONG-DROP | $z \in \{\Delta_\mu, r_\sigma\}, z < -\tau_2$ | Significant decrease |

**(2) Shape Primitive $p_{\text{shape}}$.** Nie et al. (2023) demonstrate that patch-level morphology conveys rich predictive information. The shape primitive encodes the *inter-patch trend sequence* to characterize internal evolution structure. Concretely, we partition $\mathbf{Y}_t = (y_1, \ldots, y_H)$ into $N_{\text{fcst}}$ equally-sized patches of length $L_u = H/N_{\text{fcst}}$. For each patch $i$, we compute its mean $\bar{u}_i = \frac{1}{L_u}\sum_{j=(i-1)L_u+1}^{iL_u} y_j$ and define inter-patch trend signs $s_i = \text{sgn}_\tau(\bar{u}_{i+1} - \bar{u}_i)$, where $\text{sgn}_\tau(x) = +1$ if $x > \tau$, $-1$ if $x < -\tau$, and $0$ otherwise. The dominant pattern of $(s_1, \ldots, s_{N_{\text{fcst}}-1})$ determines the shape category:

| | Category | Condition | Semantics |
|---|---|---|---|
| ╱ | ASCEND | $\forall i: s_i \geq 0 \land \exists i: s_i = +1$ | Sustained uptrend |
| ╲ | DESCEND | $\forall i: s_i \leq 0 \land \exists i: s_i = -1$ | Sustained downtrend |
| ⋀ | PEAK | rise-then-fall with sign changes | Rise then fall |
| ⋁ | TROUGH | fall-then-rise with sign changes | Fall then rise |
| ⋁⋁ | OSCILLATE | multiple sign reversals | Fluctuating |

**(3) Lag and Decay Primitive $p_{\text{lag}}$.** The same mean increase or volatility surge may exhibit vastly different response dynamics (transient pulse vs. long-tailed decay). Inspired by distributed-lag models and impulse response functions (Yang et al., 2024), we propose the lag-and-decay primitive to localize influence timing and persistence within the forecast horizon. Using the same patch partition, we compute influence intensity for each patch: $a_i = |(\bar{u}_i - \bar{X}_t)/\sigma(X_t)| + \alpha|\log((\sigma(\Delta\mathbf{u}_i) + \epsilon)/(\sigma(\Delta\mathbf{X}_t) + \epsilon))|$, and normalize to obtain an influence distribution $\pi_i = a_i/\sum_j a_j$. We then derive three indicators: (i) centroid $c = \sum_i \pi_i \cdot (i-1)/(N_{\text{fcst}}-1) \in [0,1]$, where smaller $c$ indicates earlier onset; (ii) tail mass $d = \sum_{i>i^*}\pi_i$ (with $i^* = \arg\max_i \pi_i$), where larger $d$ indicates stronger persistence; (iii) peak prominence $q = \max_i \pi_i$, distinguishing concentrated from diffuse effects. Given thresholds $\kappa_1 < \kappa_2$ (timing), $\rho$ (persistence), and $\eta$ (prominence):

| | Category | Condition | Semantics |
|---|---|---|---|
| | EARLY-FADE | $q > \eta, c \leq \kappa_1, d \leq \rho$ | Early onset, quick decay |
| | EARLY-PERSIST | $q > \eta, c \leq \kappa_1, d > \rho$ | Early onset, persistent |
| | MID-FADE | $q > \eta, \kappa_1 < c \leq \kappa_2, d \leq \rho$ | Mid-horizon pulse |
| | MID-PERSIST | $q > \eta, \kappa_1 < c \leq \kappa_2, d > \rho$ | Mid-horizon, persistent |
| | LATE | $q > \eta, c > \kappa_2$ | Late manifestation |
| | DIFFUSE | $q \leq \eta$ | Distributed impact |

The above design ensures that each primitive $p_k$ possesses *numerical verifiability*: given observation and forecast sequences, the ground-truth value $v_{t,k}$ is uniquely determined via $\psi_k$, thereby furnishing reliable supervision for the gating mechanism. All thresholds are set adaptively based on training-set statistics (e.g., quantiles).

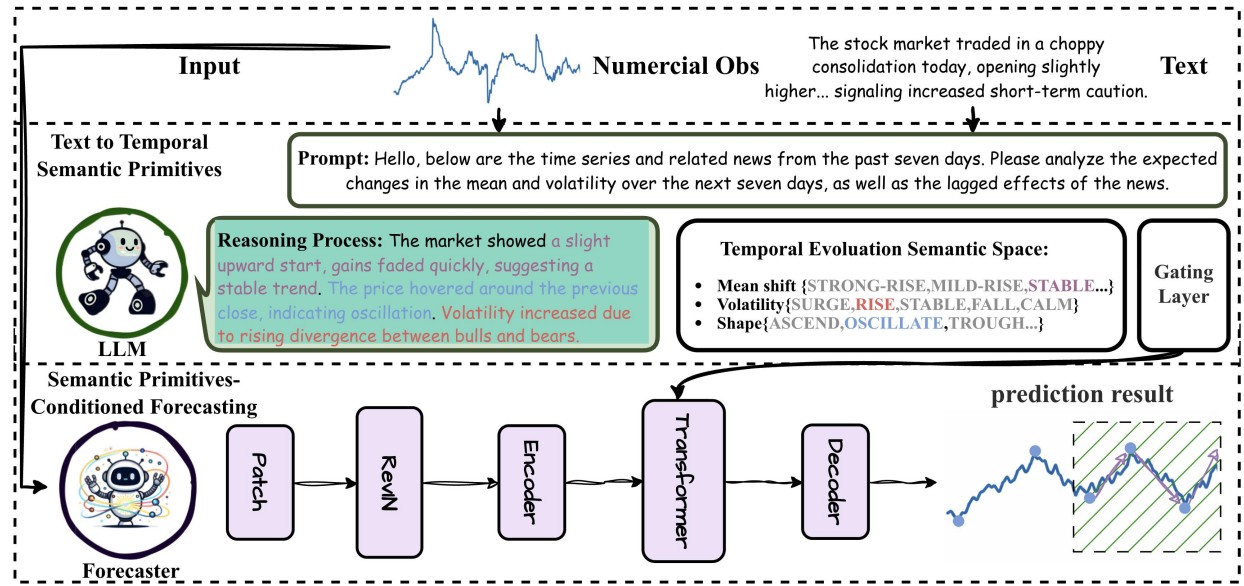

*Figure 4.* Overview of TESS. Given numerical observations and associated text, a frozen LLM extracts temporal evolution primitives (e.g., distribution shift, shape, volatility, lag) via structured prompting. These primitives, after confidence-aware gating, condition a Transformer-based forecaster that fuses semantic signals with encoded historical sequences to produce numerical predictions.

Theoretically, when the task-relevant content of the raw text can be captured by a semantic variable $V = \phi(X)$ such that $Y \perp X \mid (V, S)$ (Assumption A.1), one can restrict attention to semantic encoders that depend on $X$ only through $V$ once $S$ is observed, without losing predictive information about $Y$ conditional on $S$. In particular, Theorem A.2 shows such a restriction preserves $I(Z; Y \mid S)$ while not increasing $I(Z; X \mid S)$, and Theorem A.5 further implies this can only tighten the expected generalization bound under Assumptions A.3–A.4.

### 4.2. Text to Temporal Semantic Primitives

This stage describes how to reliably predict the discrete value of each semantic primitive $p_k$ from textual input $s_t$, and estimate its confidence to filter noisy extractions.

**LLM-based Primitive Classification.** Given the finite value domain $\mathcal{V}_k$ of each primitive, we formulate extraction as multi-class classification using a frozen LLM as the classifier. Under a structured prompt $\mathcal{D}_k$ (containing primitive semantic definitions and representative examples), we compute log-likelihood scores for each candidate $v \in \mathcal{V}_k$:

$$\ell_{t,k}(v) = \log P_{\text{LLM}}(v \mid s_t, \mathcal{D}_k). \tag{2}$$

Applying temperature-scaled softmax yields a categorical distribution and the predicted class:

$$q_{t,k}(v) = \frac{\exp(\ell_{t,k}(v)/T)}{\sum_{v'} \exp(\ell_{t,k}(v')/T)}, \tag{3}$$

$$\hat{v}_{t,k} = \arg\max_{v \in \mathcal{V}_k} q_{t,k}(v), \tag{4}$$

where $T$ is a temperature coefficient. This procedure yields both the discrete prediction $\hat{v}_{t,k}$ and the full distribution $q_{t,k}(\cdot)$, which provides calibration signals for subsequent confidence estimation.

**Confidence-Aware Gating.** Although LLMs can discriminate based on primitive definitions, real-world text often contains narrative noise and ambiguous expressions that may lead to extraction errors. To mitigate the negative impact of erroneous primitives on downstream predictions (Wang et al., 2025a), we introduce a confidence gate $g_{t,k} \in [0, 1]$ for each primitive to suppress unreliable semantic injection during inference.

Since each primitive's candidate set $\mathcal{V}_k$ is small, we adopt a simple uncertainty indicator as the calibration signal: the log-probability margin between the top-1 and top-2 candidates. Let $v^{(1)}, v^{(2)}$ denote the classes with highest and second-highest probability under $q_{t,k}(\cdot)$:

$$m_{t,k} = \log q_{t,k}(v^{(1)}) - \log q_{t,k}(v^{(2)}). \tag{5}$$

We map the predicted class $\hat{v}_{t,k}$ to an internal semantic vector $\mathbf{h}_{t,k}$ via a learnable embedding matrix $\mathbf{E}_k \in \mathbb{R}^{|\mathcal{V}_k| \times d}$:

$$\mathbf{h}_{t,k} = \mathbf{E}_k[\hat{v}_{t,k}], \tag{6}$$

where each row of $\mathbf{E}_k$ corresponds to one discrete class of primitive $k$, learned end-to-end with the downstream predictor. The gating network fuses the semantic embedding with the margin to estimate confidence:

$$g_{t,k} = \sigma\big(\mathbf{w}_k^\top [\mathbf{h}_{t,k}; \mathbf{W}_m m_{t,k}] + b_k\big), \tag{7}$$

where $\mathbf{W}_m \in \mathbb{R}^d$ projects the scalar margin to the embedding dimension. Thanks to the numerical verifiability of primitives (Section 4.1), we obtain ground-truth labels from the forecast window $v_{t,k}^* = \psi_k(\mathbf{Y}_t)$ and construct supervision labels $y_{t,k} = \mathbb{K}[\hat{v}_{t,k} = v_{t,k}^*]$. The gating network is trained with binary cross-entropy:

$$\mathcal{L}_{\text{gate}} = -\sum_{t,k}\left[y_{t,k}\log g_{t,k} + (1-y_{t,k})\log(1-g_{t,k})\right]. \quad (8)$$

During inference, we apply soft weighting $\tilde{\mathbf{h}}_{t,k} = g_{t,k} \cdot \mathbf{h}_{t,k}$, allowing high-confidence primitives to dominate predictions while suppressing interference from erroneous extractions.

Theoretically, under a mild Lipschitz assumption on the forecaster with respect to the gated primitives, the prediction error induced by an incorrect primitive is attenuated proportionally to $g_{t,k}^2$ (Theorem A.7).

### 4.3. Semantic Primitives-Conditioned Forecasting

This stage fuses numerical time series with gated semantic primitives to generate future predictions. We adopt PatchTST (Nie et al., 2023) as the backbone architecture and introduce a semantic conditioning mechanism.

**Time-Series Encoding.** Following PatchTST (Nie et al., 2023), we first apply instance normalization (Kim et al., 2022) to the input sequence $\mathbf{x} \in \mathbb{R}^L$ to mitigate distribution shift between training and testing: for each instance we compute mean $\mu$ and standard deviation $s$, then normalize before patching. The input is segmented into $N = \lfloor(L - P)/S\rfloor + 1$ patches, where $P$ is the patch length and $S$ is the stride. Each patch is mapped to the $d$-dimensional latent space via a trainable linear projection $\mathbf{W}_p \in \mathbb{R}^{d \times P}$, with learnable positional encodings $\mathbf{W}_{\text{pos}} \in \mathbb{R}^{d \times N}$ added to obtain patch embeddings $\mathbf{E}_{\text{patch}} \in \mathbb{R}^{N \times d}$.

**Semantic Primitives-Conditioned Prediction.** The $K$ gated semantic vectors $\tilde{\mathbf{h}}_{t,k} \in \mathbb{R}^d$ from Section 4.2 are stacked to form semantic prefix tokens $\mathbf{P} \in \mathbb{R}^{K \times d}$, concatenated with patch embeddings to form the Transformer input:

$$\mathbf{Z}^{(0)} = [\mathbf{P}; \mathbf{E}_{\text{patch}}] \in \mathbb{R}^{(K+N) \times d}. \quad (9)$$

The input sequence is processed by $M$ Transformer encoder layers. Each layer first captures sequential dependencies via $H$-head self-attention:

$$\text{Attn}(\mathbf{Q}, \mathbf{K}, \mathbf{V}) = \text{Softmax}\left(\frac{\mathbf{Q}\mathbf{K}^\top}{\sqrt{d/H}}\right)\mathbf{V}, \quad (10)$$

where $\mathbf{Q}, \mathbf{K}, \mathbf{V} \in \mathbb{R}^{(K+N) \times (d/H)}$ are obtained via linear projections. Multi-head outputs are concatenated and updated through feed-forward networks with residual connections. This prefix fusion allows semantic information to

participate in temporal modeling throughout the attention mechanism. Finally, we extract patch-corresponding outputs $\mathbf{Z}_{\text{out}} \in \mathbb{R}^{N \times d}$, flatten and map to the forecast horizon via an MLP, and apply inverse normalization to recover the original scale:

$$\hat{\mathbf{y}} = s \cdot \text{MLP}(\text{Flatten}(\mathbf{Z}_{\text{out}})) + \mu \in \mathbb{R}^H. \quad (11)$$

**Training Objective.** The model is trained end-to-end (with the LLM frozen) using MSE loss to measure prediction-target discrepancy, jointly optimized with gating supervision:

$$\mathcal{L} = \mathcal{L}_{\text{fcst}} + \lambda\mathcal{L}_{\text{gate}}, \quad \mathcal{L}_{\text{fcst}} = \frac{1}{H}\|\hat{\mathbf{y}} - \mathbf{y}\|_2^2. \quad (12)$$

## 5. Experiments

### 5.1. Experimental Setup

**Benchmark.** We evaluate TESS on four real-world datasets spanning financial and general domains. **Financial time-series datasets** include FNSPID (Dong et al., 2024) and Bitcoin, both exhibiting significant event-driven non-stationarity. **General time-series datasets** include Electricity and Environment, used to verify cross-domain generalization. For all datasets, we strictly follow the standard evaluation protocols and official data splits from the original literature to ensure fair comparison and reproducibility (Qiu et al., 2024; Shao et al., 2024). Detailed dataset statistics are provided in Appendix B.

**Baselines.** We compare TESS against two categories of baselines: (i) **Unimodal time-series models**, including TimeMixer (Wang et al., 2024b), TSMixer (Chen et al., 2023), Nonstationary Transformer (Liu et al., 2022b), TimesNet (Wu et al., 2023), FEDformer (Zhou et al., 2022), Pyraformer (Liu et al., 2022a), Reformer (Kitaev et al., 2020), and PatchTST (Nie et al., 2023), covering mainstream architectures such as MLP and Transformer variants, all implemented using Time-Series-Library[1]; (ii) **Multimodal fusion models**, including TimeLLM (Jin et al., 2024), ChatTime (Wang et al., 2025b), and NewsForecasting (Wang et al., 2024a), all using official implementations. Hyperparameters for all baselines are determined via grid search on the validation set to ensure fair comparison.

**Implementation Details.** All experiments are implemented in PyTorch (Paszke et al., 2019) and conducted on 8 NVIDIA A800 80GB GPUs. We use Qwen3-8B for primitive extraction in the main experiments. GPT-5.2 is used only to generate semi-synthetic diagnostic text in Section 3.

---

[1] https://github.com/thuml/Time-Series-Library

*Table 1.* Comparison of baselines for time-series forecasting. Lower scores indicate better performance. Bold red indicates the best result, and bold blue indicates the runner-up.

| Model | Bitcoin (Finance) | | | FNSPID (Finance) | | | Electricity (General) | | | Environment (General) | | |
|---|---|---|---|---|---|---|---|---|---|---|---|---|
| | MAE | MSE | RMSE | MAE | MSE | RMSE | MAE | MSE | RMSE | MAE | MSE | RMSE |
| **Traditional Models** | | | | | | | | | | | | |
| TimeMixer | 1.6757 | 4.3725 | 2.0910 | 0.0153 | 0.0017 | 0.0407 | 0.1064 | 0.0252 | 0.1586 | 0.4219 | 0.3693 | 0.6077 |
| TSMixer | 4.1202 | 29.1934 | 5.4031 | 0.0353 | 0.0082 | 0.0907 | 0.1119 | 0.0258 | 0.1606 | 0.4860 | 0.3878 | 0.6227 |
| Nonstationary | 1.6003 | 4.3420 | 2.0837 | 0.0154 | 0.0016 | 0.0400 | 0.1066 | 0.0256 | 0.1600 | **0.4178** | **0.3472** | **0.5892** |
| TimesNet | 1.4598 | 3.8229 | 1.9552 | **0.0152** | **0.0015** | 0.0385 | 0.1035 | **0.0242** | **0.1557** | 0.4258 | 0.3473 | **0.5893** |
| FEDformer | 1.4197 | 3.2768 | 1.8102 | 0.0163 | 0.0016 | 0.0396 | 0.1144 | 0.0267 | 0.1633 | 0.4513 | 0.3650 | 0.5893 |
| Pyraformer | 1.3994 | 3.6377 | 1.9073 | 0.0168 | 0.0015 | **0.0383** | 0.1117 | 0.0261 | 0.1616 | 0.4303 | 0.3649 | 0.6041 |
| Reformer | 12.4630 | 157.8843 | 12.5652 | 0.0299 | 0.0141 | 0.1188 | 0.1180 | 0.0292 | 0.1709 | 0.4258 | 0.3473 | 0.5893 |
| PatchTST | 1.3615 | 3.2456 | 1.8016 | 0.0160 | 0.0016 | 0.0402 | **0.1047** | 0.0243 | 0.1559 | 0.4290 | 0.3706 | 0.6088 |
| **Multimodal Models** | | | | | | | | | | | | |
| TimeLLM | 1.3940 | 3.4477 | 1.8568 | 0.0158 | 0.0017 | 0.0412 | 0.1074 | 0.0255 | 0.1596 | 0.4237 | 0.3714 | 0.6094 |
| ChatTime | 1.4774 | 3.9389 | 1.9846 | 0.0187 | 0.0018 | 0.0424 | 0.1149 | 0.0279 | 0.1670 | 0.4344 | 0.3787 | 0.6153 |
| NewsForecasting | **1.3593** | **3.2036** | **1.7898** | 0.0176 | 0.0017 | 0.0412 | 0.1088 | 0.0254 | 0.1593 | 0.4378 | 0.3678 | 0.6064 |
| Ours | **1.1120** | **2.2726** | **1.5075** | **0.0147** | **0.0012** | **0.0347** | **0.1031** | **0.0230** | **0.1518** | **0.4202** | **0.3534** | 0.5945 |
| *Gain vs. best (%)* | +18.2% | +29.1% | +15.8% | +3.3% | +20.0% | +9.9% | +0.4% | +5.0% | +2.5% | −0.6% | −1.8% | −0.9% |

*Table 2.* Ablation study on the contributions of temporal evolution semantic space (TESS) and gating mechanism. Lower scores indicate better performance. Bold red indicates the best result.

| Dataset | Metric | w/o TESS | w/o Gating | TESS |
|---|---|---|---|---|
| **Bitcoin** | MAE | 1.6025 | 1.1532 | **1.1120** |
| | MSE | 4.2238 | 2.3556 | **2.2726** |
| | RMSE | 2.0552 | 1.5349 | **1.5075** |
| **FNSPID** | MAE | 0.0155 | 0.0151 | **0.0147** |
| | MSE | 0.0018 | 0.0015 | **0.0012** |
| | RMSE | 0.0412 | 0.0387 | **0.0347** |
| **Electricity** | MAE | 0.1214 | 0.1070 | **0.1031** |
| | MSE | 0.0298 | 0.0248 | **0.0230** |
| | RMSE | 0.1726 | 0.1575 | **0.1518** |

Primitive extraction is performed offline as a preprocessing step, after which the online forecaster consumes only compact primitive labels. We use the AdamW optimizer (Loshchilov & Hutter, 2017) with learning rate selected from $\{1e{-}4, 5e{-}4, 1e{-}3\}$. Training employs early stopping (patience=10), terminating when validation loss ceases to decrease. Following the settings in TFB (Qiu et al., 2024), we do not apply the "Drop Last" trick to ensure a fair comparison.

**5.2. Main Results**

Table 1 presents the performance comparison between TESS and all baselines across four datasets. (i) **Significant improvements on financial datasets.** On financial datasets exhibiting pronounced non-stationarity (FNSPID and Bitcoin), TESS achieves substantial performance gains. Specifically, on FNSPID, TESS improves over the strongest baseline TimesNet by 3.3%, 20.0%, and 9.9% across three metrics. On Bitcoin, TESS outperforms the strongest baseline NewsForecasting by 18.2%, 29.1%, and 15.8% in terms of MAE, MSE, and RMSE, respectively. These results demonstrate the superiority of the temporal evolution semantic space in capturing event-driven non-stationary dynamics. (ii) **Stable performance on general datasets.** On general datasets, TESS likewise demonstrates competitive performance. On Electricity, TESS achieves the best performance across all metrics, improving over the strongest baseline by 0.4%–5.0%. On Environment, TESS attains runner-up performance with less than 1% gap from the strongest baseline Nonstationary Transformer. These results indicate that TESS maintains stable predictive capability even in scenarios with relatively mild non-stationarity.

**Effectiveness in Non-Stationary Scenarios.** To further validate TESS's performance under different non-stationarity patterns, we extract three subsets from the test set, each representing a non-stationary scenario: (i) **Shape transition**, (ii) **Volatility change**, and (iii) **Distribution shift**. Figure 5 presents the MSE performance across different methods. TESS achieves consistent improvements across all three non-stationary scenarios, with MSE reductions of 21–52% over multimodal baselines and 21–45% over unimodal baselines on FNSPID and Bitcoin.

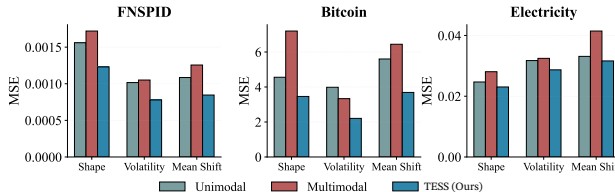

*Figure 5.* Performance comparison on three types of non-stationary scenarios (Shape, Volatility, Distribution Shift) across three datasets. TESS (blue) consistently outperforms both unimodal (teal) and multimodal (red) baselines.

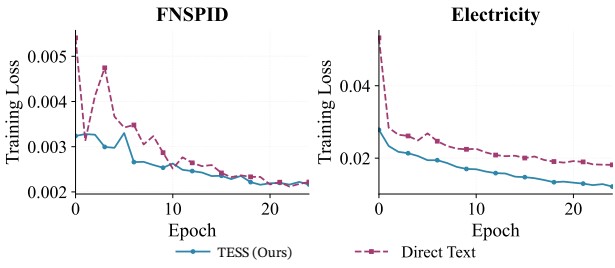

*Figure 6.* Training loss curves on FNSPID and Electricity datasets. TESS (blue) converges faster with smoother dynamics than direct text fusion (purple).

## 5.3. Model Analysis

**Ablation Study.** We ablate TESS and the gating mechanism on three datasets. Since gating relies on primitives extracted by TESS, removing TESS necessarily entails removing gating as well. Table 2 presents the results. Removing TESS leads to substantial degradation, with MSE increasing by 46.2%, 29.4%, and 22.8% respectively, confirming its effectiveness. In contrast, removing gating incurs more modest increases of 3.7%, 2.6%, and 7.5%, indicating that confidence-aware gating effectively filters extraction errors. These results underscore the necessity of both components: TESS drives performance gains, while gating enhances robustness against LLM extraction errors.

**Case Study.** Figure 7 illustrates four representative prediction cases. On Bitcoin, TESS successfully captures distribution shift and trend patterns (e.g., MILD-RISE, TROUGH); on Electricity, it identifies shape transitions with appropriate temporal localization (e.g., MID-FADE, LATE). In contrast, direct text fusion fails to capture these patterns, leading to substantial prediction deviations. These results confirm that the semantic space effectively bridges implicit textual signals and quantifiable forecasting gains.

**Effectiveness of Temporal Evolution Semantic Space.** We examine training dynamics and primitive ablation. First, Figure 6 shows that TESS exhibits faster and smoother

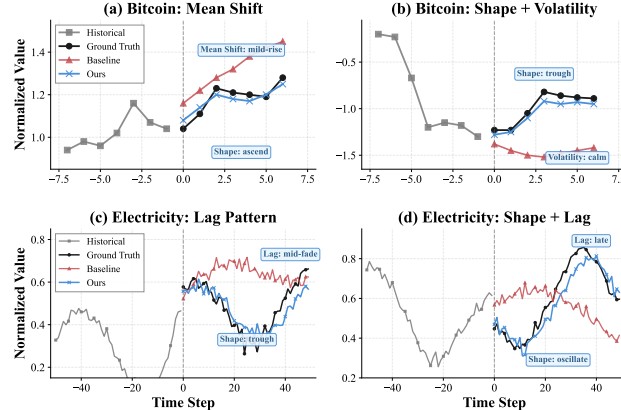

*Figure 7.* Case study on Bitcoin and Electricity datasets. TESS (blue) accurately captures temporal evolution primitives and aligns closely with ground truth (black), while the baseline (red) fails to capture critical patterns.

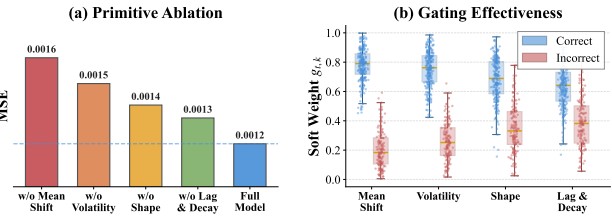

*Figure 8.* Analysis of temporal evolution semantic space. **(a)** Primitive ablation on FNSPID: removing each primitive leads to MSE increase, with Distribution Shift having the largest impact. **(b)** Gating effectiveness: correctly extracted primitives (blue) receive significantly higher soft weights than incorrect ones (red).

convergence than direct text fusion on both FNSPID and Electricity, indicating that the semantic space alleviates cross-modal optimization challenges. Second, Figure 8(a) presents primitive ablation on FNSPID: removing Distribution Shift alone increases MSE by 33%, underscoring its role in capturing non-stationarity; removing other primitives also incurs performance loss. Best performance is achieved when all four primitives are jointly used, confirming their complementarity.

**Effectiveness of Gating Mechanism.** To validate the effectiveness of the confidence-aware gating mechanism, we analyze the relationship between gating soft weights and primitive extraction correctness. Figure 8(b) presents the soft weight distributions across the four primitives on the FNSPID dataset. The results demonstrate that gating weights are highly correlated with primitive correctness: correctly extracted samples exhibit median weights predominantly in the range of 0.65–0.78, whereas incorrectly extracted samples show median weights of only 0.21–0.40. This indicates that the gating network successfully learns

to map the uncertainty signals from LLM outputs to reliability estimates, effectively suppressing the interference of erroneous primitives on predictions through soft weighting.

# 6. Conclusion

This paper proposes TESS, a two-stage framework that bridges the modality gap by projecting text into a Temporal Evolution Semantic Space of interpretable primitives before conditioning numerical forecasters. A confidence-aware gating layer further improves robustness against LLM extraction errors. Extensive experiments across financial and general domains demonstrate that our proposed TESS achieves up to 29% MSE reduction over state-of-the-art baselines.

# 7. Limitations

The current semantic space is also intentionally limited to four fixed primitives for verifiability and interpretability. These primitives may miss signals such as seasonality, periodicity, or fine-grained causal effects, so the semantic sufficiency assumption is scoped rather than universal. The semi-synthetic diagnostic is likewise limited in domain and LLM coverage. Future work may introduce more numerically verifiable primitives or learn task-specific primitives from data. These extensions can be incorporated into the current extraction, gating, and forecasting stages. It is also worth noting that existing multimodal time-series datasets remain relatively small; consequently, both baselines and our method may exhibit some fluctuations across different GPU types, but the relative performance gaps remain consistent, and TESS still achieves state-of-the-art performance.

# Acknowledgements

This work was supported in part by the following grants and programs: the National Natural Science Foundation of China (No. 62476154); the Major Basic Research Project of Shandong Provincial Natural Science Foundation (No. ZR2024ZD03); the Natural Science Foundation of Shandong Province (No. ZR2024MF101); the Shandong University Qilu Young Scholars Program; the Excellent Young Scientists Fund Program (Overseas) of the Shandong Provincial Natural Science Foundation (No. 2026HWYQ-007); the Youth Innovation Science and Technology Support Program for Higher Education Institutions of Shandong Province (No. 2025KJH105); the Young Expert of Taishan Scholars Program (No. tsqn202312026); and the Open Project Program of State Key Laboratory of Virtual Reality Technology and Systems, Beihang University (No. VRLAB2024A02).

# Impact Statement

This paper presents work aimed at advancing time-series analysis and multimodal forecasting. The proposed method may support applications in finance, energy management, and environmental monitoring by improving forecasting under event-driven non-stationarity. However, forecasting systems may affect downstream decisions, and inaccurate predictions can lead to financial or operational risks if deployed without adequate validation. We therefore encourage practitioners to consider data quality, model uncertainty, and domain-specific constraints when applying the proposed method in real-world settings.

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

# A. Theorems & Proofs

We study representation learning from raw text $X$ in the presence of an observed time series $S$ and a prediction target $Y$. Let $V = \phi(X)$ denote a deterministic semantic compression of $X$, taking values in a finite or countable set $\mathcal{V}$. Our analysis relies on the following sufficiency assumption.

**Assumption A.1** (Semantic sufficiency). The target $Y$ is conditionally independent of the raw text $X$ given the semantic representation $V$ and the time series $S$, i.e.,

$$Y \perp X \mid (V, S), \qquad \text{equivalently } p(y \mid x, s) = p(y \mid v, s) \text{ for } v = \phi(x). \tag{13}$$

We now characterize the implications of Assumption A.1 for the design of text encoders.

**Theorem A.2** (Information-preserving restriction to semantic encoders). *Let $p(z \mid x)$ be an arbitrary encoder mapping the raw text $X$ to a representation $Z \in \mathcal{Z}$, and define the induced joint distribution*

$$p(x, s, y, z) = p(z \mid x) \, p(y \mid x, s) \, p(x, s). \tag{14}$$

*Under Assumption A.1, there exists a context-conditioned encoder $\tilde{p}(z \mid x, s)$ of the form*

$$\tilde{p}(z \mid x, s) = q(z \mid v, s), \qquad v = \phi(x),$$

*such that*

$$I_{\tilde{p}}(Z; Y \mid S) = I_p(Z; Y \mid S), \tag{15}$$
$$I_{\tilde{p}}(Z; X \mid S) \le I_p(Z; X \mid S). \tag{16}$$

*Proof.* We denote by $p$ both the data distribution $p(x, s, y)$ and the original encoder distribution $p(z \mid x)$, with the understanding that the factorization of the joint over $(X, S, Y, Z)$ is

$$p(x, s, y, z) = p(z \mid x) \, p(y \mid x, s) \, p(x, s), \tag{17}$$

which follows from the chain rule and the assumption that $Z$ is generated from $X$ only, i.e., $Z \perp (Y, S) \mid X$. The relevant conditional mutual information terms are

$$I_p(Z; Y \mid S) := \sum_s p(s) \sum_{z,y} p(z, y \mid s) \log \frac{p(z, y \mid s)}{p(z \mid s) \, p(y \mid s)}, \tag{18}$$

$$I_p(Z; X \mid S) := \sum_s p(s) \sum_x p(x \mid s) \sum_z p(z \mid x) \log \frac{p(z \mid x)}{p(z \mid s)}. \tag{19}$$

Our goal is to construct a new encoder $\tilde{p}(z \mid x, s)$ and show that (15)–(16) hold.

Let $V = \phi(X)$ and define

$$p(v, s) := \sum_{x:\phi(x)=v} p(x, s), \tag{20}$$

and, for all $x$ such that $\phi(x) = v$,

$$\alpha_{x|v,s} := \frac{p(x, s)}{p(v, s)}. \tag{21}$$

Then $\sum_{x:\phi(x)=v} \alpha_{x|v,s} = 1$ and hence

$$p(z \mid v, s) := \sum_{x:\phi(x)=v} \alpha_{x|v,s} \, p(z \mid x), \tag{22}$$

$$\sum_z p(z \mid v, s) = \sum_{x:\phi(x)=v} \alpha_{x|v,s} \sum_z p(z \mid x) = \sum_{x:\phi(x)=v} \alpha_{x|v,s} = 1,$$

so $p(z \mid v, s)$ is a valid conditional distribution. Define the new encoder

$$\tilde{p}(z \mid x, s) := p(z \mid v, s), \qquad v = \phi(x), \tag{23}$$

which depends on $x$ only through $v = \phi(x)$ once the observed time series $s$ is given. We next prove (15) and (16).

Starting from (18), it suffices to show that the induced joint distribution over $(Z, Y, S)$ is unchanged under $\tilde{p}$, since then the corresponding conditionals given $S$ coincide and the value of $I(Z; Y \mid S)$ is preserved:

$$p_{\tilde{p}}(z, y, s) = p(z, y, s) \ \forall z, y, s \implies I_{\tilde{p}}(Z; Y \mid S) = I_p(Z; Y \mid S). \tag{24}$$

Under the original encoder,

$$p(z, y, s) = \sum_x p(z \mid x) \, p(y \mid x, s) \, p(x, s) = \sum_x p(z \mid x) \, p(y \mid v, s) \, p(x, s) \tag{25}$$

$$= \sum_v \sum_{x:\phi(x)=v} p(z \mid x) \, p(y \mid v, s) \, p(v, s) \, \alpha_{x \mid v, s} \tag{26}$$

$$= \sum_v p(y \mid v, s) \, p(v, s) \sum_{x:\phi(x)=v} \alpha_{x \mid v, s} \, p(z \mid x) = \sum_v p(y \mid v, s) \, p(v, s) \, p(z \mid v, s), \tag{27}$$

where (25) uses (13), (26) uses (20)–(21), and (27) uses (22). Under the new encoder $\tilde{p}(z \mid x, s) = p(z \mid v, s)$,

$$p_{\tilde{p}}(z, y, s) = \sum_x \tilde{p}(z \mid x, s) \, p(y \mid x, s) \, p(x, s) = \sum_x p(z \mid v, s) \, p(y \mid v, s) \, p(x, s) \tag{28}$$

$$= \sum_v p(z \mid v, s) \, p(y \mid v, s) \sum_{x:\phi(x)=v} p(x, s) = \sum_v p(z \mid v, s) \, p(y \mid v, s) \, p(v, s), \tag{29}$$

using again (13) and (20). Comparing (27) and (29) yields $p_{\tilde{p}}(z, y, s) = p(z, y, s)$ for all $z, y, s$, and hence for all $s$,

$$p_{\tilde{p}}(z, y \mid s) = p(z, y \mid s), \qquad p_{\tilde{p}}(z \mid s) = p(z \mid s), \qquad p_{\tilde{p}}(y \mid s) = p(y \mid s).$$

Substituting these equalities into (18) gives $I_{\tilde{p}}(Z; Y \mid S) = I_p(Z; Y \mid S)$, proving (15).

We prove (16). For the original encoder, $p(z \mid s) = \sum_x p(z \mid x) p(x \mid s)$. Using $p(x, s) = p(v, s) \alpha_{x \mid v, s}$ for $v = \phi(x)$, we can group terms as

$$I_p(Z; X \mid S) = \sum_{v, s} p(v, s) \sum_{x:\phi(x)=v} \alpha_{x \mid v, s} \sum_z p(z \mid x) \log \frac{p(z \mid x)}{p(z \mid s)}. \tag{30}$$

Under $\tilde{p}(z \mid x, s) = p(z \mid v, s)$, the conditional marginal of $Z$ given $s$ is unchanged:

$$p_{\tilde{p}}(z \mid s) = \sum_v p(v \mid s) p(z \mid v, s) = \sum_v p(v \mid s) \sum_{x:\phi(x)=v} \alpha_{x \mid v, s} p(z \mid x) = p(z \mid s). \tag{31}$$

Hence the denominator $p(z \mid s)$ is the same for $p$ and $\tilde{p}$, and

$$I_{\tilde{p}}(Z; X \mid S) = \sum_{v, s} p(v, s) \sum_z p(z \mid v, s) \log \frac{p(z \mid v, s)}{p(z \mid s)}. \tag{32}$$

Define $f_s(r) := \sum_z r(z) \log \frac{r(z)}{p(z \mid s)} = D_{\mathrm{KL}}(r \| p(z \mid s))$. By convexity of $D_{\mathrm{KL}}(\cdot \| p(z \mid s))$ in its first argument, for each fixed $(v, s)$,

$$\sum_z p(z \mid v, s) \log \frac{p(z \mid v, s)}{p(z \mid s)} = f_s \left( \sum_{x:\phi(x)=v} \alpha_{x \mid v, s} p(\cdot \mid x) \right) \leq \sum_{x:\phi(x)=v} \alpha_{x \mid v, s} f_s(p(\cdot \mid x))$$

$$= \sum_{x:\phi(x)=v} \alpha_{x \mid v, s} \sum_z p(z \mid x) \log \frac{p(z \mid x)}{p(z \mid s)}. \tag{33}$$

Multiplying (33) by $p(v, s)$ and summing over $(v, s)$ yields $I_{\tilde{p}}(Z; X \mid S) \leq I_p(Z; X \mid S)$, which proves (16). Together with (15), this completes the proof.

$\square$

We next show that the semantic restriction in Theorem A.2 yields a direct improvement in information-theoretic generalization bounds, following the mutual-information framework of Xu & Raginsky (2017).

**Setup.** Let $\mathcal{S} = \{(X_i, S_i, Y_i)\}_{i=1}^n \sim p(x, s, y)^{\otimes n}$ be the training set. Given an encoder $p(z \mid x)$, let $Z_i \sim p(\cdot \mid X_i)$ independently and write $Z = (Z_1, \ldots, Z_n)$. Define the population and empirical risks under loss $\ell$ by

$$L_{\text{pop}}(Z) := \mathbb{E}_{(X,S,Y)}[\ell(Z, Y)], \qquad L_{\text{emp}}(Z) := \frac{1}{n} \sum_{i=1}^n \ell(Z_i, Y_i).$$

**Assumption A.3** (Sub-Gaussian loss). For every $(x, y)$, the random variable $\ell(Z, y)$ is $\sigma$-sub-Gaussian under $Z \sim p(\cdot \mid x)$, i.e.,

$$\mathbb{E}\big[\exp\big(\lambda(\ell(Z, y) - \mathbb{E}[\ell(Z, y)])\big)\big] \leq \exp\left(\frac{\sigma^2 \lambda^2}{2}\right), \quad \forall \lambda \in \mathbb{R}.$$

**Assumption A.4** (Conditional independence of samples). Conditioned on $\mathcal{S}$, the encoder acts independently across samples: $Z_i \perp Z_j$ for $i \neq j$. Consequently,

$$I(Z; \mathcal{S}) = \sum_{i=1}^n I(Z_i; X_i \mid S_i) = n\, I(Z; X \mid S).$$

**Theorem A.5** (Generalization under semantic compression). *Let $p(z \mid x)$ be any encoder and let $\tilde{p}(z \mid x, s)$ be the semantic encoder constructed in Theorem A.2. Under Assumptions A.3–A.4, the expected generalization gaps satisfy*

$$\text{Gen}(p) := |\mathbb{E}[L_{\text{pop}}(Z) - L_{\text{emp}}(Z)]| \leq \sqrt{\frac{2\sigma^2}{n} I_p(Z; X \mid S)}, \qquad \text{Gen}(\tilde{p}) \leq \sqrt{\frac{2\sigma^2}{n} I_{\tilde{p}}(Z; X \mid S)}.$$

*Moreover, since Theorem A.2 guarantees $I_{\tilde{p}}(Z; X \mid S) \leq I_p(Z; X \mid S)$, it follows that*

$$\text{Gen}(\tilde{p}) \leq \text{Gen}(p).$$

*Proof.* We follow the information-theoretic generalization framework of Xu & Raginsky (2017). By Assumption A.3, Theorem 1 of Xu & Raginsky (2017) gives

$$|\mathbb{E}[L_{\text{pop}}(Z) - L_{\text{emp}}(Z)]| \leq \sqrt{\frac{2\sigma^2}{n} I(Z; \mathcal{S})}. \tag{34}$$

By Assumption A.4,

$$I(Z; \mathcal{S}) = \sum_{i=1}^n I(Z_i; X_i \mid S_i) = n\, I(Z; X \mid S). \tag{35}$$

Combining (34)–(35) yields

$$\text{Gen}(p) = |\mathbb{E}[L_{\text{pop}}(Z) - L_{\text{emp}}(Z)]| \leq \sqrt{\frac{2\sigma^2}{n} I_p(Z; X \mid S)}. \tag{36}$$

Applying the same argument to $\tilde{p}$ gives

$$\text{Gen}(\tilde{p}) \leq \sqrt{\frac{2\sigma^2}{n} I_{\tilde{p}}(Z; X \mid S)}. \tag{37}$$

Finally, Theorem A.2 implies $I_{\tilde{p}}(Z; X \mid S) \leq I_p(Z; X \mid S)$, hence

$$\text{Gen}(\tilde{p}) \leq \text{Gen}(p).$$

$\square$

**Assumption A.6** (Lipschitz dependence). Fix a forecasting time $t$ and let

$$\hat{Y}_t \;=\; F\big(E_{\text{time},t},\,\tilde{h}_1,\ldots,\tilde{h}_K\big), \qquad \tilde{h}_k = g_{t,k} h_k, \;\; g_{t,k} \in [0,1].$$

Assume that for each $k \in [K]$, the map $a \mapsto F(E_{\text{time},t}, \tilde{h}_1,\ldots,\tilde{h}_{k-1}, a, \tilde{h}_{k+1},\ldots,\tilde{h}_K)$ is $L_k$-Lipschitz w.r.t. $\|\cdot\|$:

$$\big|F(\ldots,a,\ldots) - F(\ldots,b,\ldots)\big| \leq L_k \|a - b\|, \qquad \forall\, a,b \in \mathbb{R}^d.$$

**Theorem A.7** (stability). *Under Assumption A.6, let $\tilde{h}_k^{\text{true}} = g_{t,k} h_k^{\text{true}}$ and $\tilde{h}_k^{\text{err}} = g_{t,k} h_k^{\text{err}}$, and define $\Delta_k := \|h_k^{\text{err}} - h_k^{\text{true}}\|$. Denote the corresponding predictions by*

$$\hat{Y}_t^{\text{true}} := F(E_{\text{time},t}, \tilde{h}_1^{\text{true}},\ldots,\tilde{h}_K^{\text{true}}), \qquad \hat{Y}_t^{\text{err}} := F(E_{\text{time},t}, \tilde{h}_1^{\text{err}},\ldots,\tilde{h}_K^{\text{err}}).$$

*Then*

$$\big(\hat{Y}_t^{\text{err}} - \hat{Y}_t^{\text{true}}\big)^2 \;\leq\; K \sum_{k=1}^{K} L_k^2\, g_{t,k}^2\, \Delta_k^2.$$

*Proof.* Let $E := E_{\text{time},t}$ and $\Delta_k := \|h_k^{\text{err}} - h_k^{\text{true}}\|$. Define $\mathbf{h}^{\text{true}} := (g_{t,1} h_1^{\text{true}},\ldots,g_{t,K} h_K^{\text{true}})$ and $\mathbf{h}^{\text{err}} := (g_{t,1} h_1^{\text{err}},\ldots,g_{t,K} h_K^{\text{err}})$. For $k \in \{0,1,\ldots,K\}$, set

$$\mathbf{z}^{(k)} := \big(g_{t,1} h_1^{\text{err}},\ldots,g_{t,k} h_k^{\text{err}}, g_{t,k+1} h_{k+1}^{\text{true}},\ldots,g_{t,K} h_K^{\text{true}}\big),$$

so that $\mathbf{z}^{(0)} = \mathbf{h}^{\text{true}}$ and $\mathbf{z}^{(K)} = \mathbf{h}^{\text{err}}$. Then, by telescoping, the triangle inequality, Assumption A.6, and $\|g_{t,k}(h_k^{\text{err}} - h_k^{\text{true}})\| = g_{t,k}\Delta_k$,

$$\big|\hat{Y}_t^{\text{err}} - \hat{Y}_t^{\text{true}}\big| = \big|F(E,\mathbf{z}^{(K)}) - F(E,\mathbf{z}^{(0)})\big| \leq \sum_{k=1}^{K} \big|F(E,\mathbf{z}^{(k)}) - F(E,\mathbf{z}^{(k-1)})\big|$$

$$\leq \sum_{k=1}^{K} L_k \big\|g_{t,k} h_k^{\text{err}} - g_{t,k} h_k^{\text{true}}\big\| = \sum_{k=1}^{K} L_k\, g_{t,k}\, \Delta_k.$$

Finally, applying $(\sum_{k=1}^{K} a_k)^2 \leq K \sum_{k=1}^{K} a_k^2$ with $a_k := L_k g_{t,k} \Delta_k$ yields

$$\big(\hat{Y}_t^{\text{err}} - \hat{Y}_t^{\text{true}}\big)^2 \leq K \sum_{k=1}^{K} L_k^2\, g_{t,k}^2\, \Delta_k^2,$$

as claimed. $\square$

**Setup.** Let $V = \phi(X) \in \mathcal{V}$ denote the TESS primitives, where

$$\mathcal{V} = \mathcal{V}_1 \times \cdots \times \mathcal{V}_K, \qquad |\mathcal{V}_k| = M_k, \qquad |\mathcal{V}| = M := \prod_{k=1}^{K} M_k.$$

Consider predictors of the form $\hat{Y} = g(V)$ with $g : \mathcal{V} \to [-1,1]$. Let $\ell(\hat{y},y) = (\hat{y} - y)^2 \in [0,4]$ be the squared loss, and define

$$R(g) = \mathbb{E}\,\ell(g(V), Y), \qquad \widehat{R}_n(g) = \frac{1}{n} \sum_{i=1}^{n} \ell(g(V_i), Y_i).$$

**Theorem A.8.** *There exists a universal constant $C > 0$ such that for any $\delta \in (0,1)$, with probability at least $1 - \delta$ over $n$ i.i.d. samples,*

$$\sup_{g:\mathcal{V} \to [-1,1]} \big|R(g) - \widehat{R}_n(g)\big| \;\leq\; C\left(\sqrt{\frac{M}{n}} + \sqrt{\frac{\log(1/\delta)}{n}}\right).$$

*Consequently, it suffices that $n \gtrsim M/\varepsilon^2$ to guarantee $\sup_g |R(g) - \widehat{R}_n(g)| \leq \varepsilon$.*

*Proof.* Define $\mathcal{G} := \{g : \mathcal{V} \to [-1, 1]\}$ and its empirical Rademacher complexity

$$\widehat{\mathfrak{R}}_n(\mathcal{G}) = \mathbb{E}_\sigma \left[ \sup_{g \in \mathcal{G}} \frac{1}{n} \sum_{i=1}^n \sigma_i g(V_i) \right], \qquad \sigma_i \stackrel{iid}{\sim} \{\pm 1\}.$$

Group samples by primitive value and let $N_v := |\{i : V_i = v\}|$ so that $\sum_{v \in \mathcal{V}} N_v = n$. Then

$$\sup_{g \in \mathcal{G}} \sum_{i=1}^n \sigma_i g(V_i) = \sup_g \sum_{v \in \mathcal{V}} g(v) \sum_{i:V_i=v} \sigma_i = \sum_{v \in \mathcal{V}} \Big| \sum_{i:V_i=v} \sigma_i \Big|.$$

Taking expectation over $\sigma$ and using $\mathbb{E}_\sigma |\sum_{j=1}^m \sigma_j| \le \sqrt{m}$ yields

$$\widehat{\mathfrak{R}}_n(\mathcal{G}) \le \frac{1}{n} \sum_{v \in \mathcal{V}} \sqrt{N_v}.$$

By Cauchy–Schwarz,

$$\sum_{v \in \mathcal{V}} \sqrt{N_v} \le \sqrt{\Big( \sum_v 1 \Big) \Big( \sum_v N_v \Big)} = \sqrt{Mn},$$

hence $\widehat{\mathfrak{R}}_n(\mathcal{G}) \le \sqrt{M/n}$.

Since $\ell(\cdot, y)$ is 4-Lipschitz on $[-1, 1]$, the contraction inequality gives

$$\widehat{\mathfrak{R}}_n(\ell \circ \mathcal{G}) \le 4\,\widehat{\mathfrak{R}}_n(\mathcal{G}) \le 4\sqrt{\frac{M}{n}}.$$

Applying the standard Rademacher generalization bound (e.g. Bartlett & Mendelson, 2002), with probability at least $1 - \delta$,

$$\sup_{g \in \mathcal{G}} \big( R(g) - \widehat{R}_n(g) \big) \le 2\,\widehat{\mathfrak{R}}_n(\ell \circ \mathcal{G}) + c\sqrt{\frac{\log(1/\delta)}{n}} \le C\left( \sqrt{\frac{M}{n}} + \sqrt{\frac{\log(1/\delta)}{n}} \right),$$

for universal constants $c, C > 0$. The same bound holds for the absolute deviation. $\qquad\square$

*Remark* A.9. Theorem A.8 yields a complexity term $\asymp \sqrt{M/n}$ with $M = \prod_{k=1}^K M_k$. For a token-level combinatorial search over a finite set $\mathcal{A}_T$, a standard finite-class bound incurs a term $\asymp \sqrt{\log |\mathcal{A}_T|/n}$. Thus the reduction is captured by

$$\sqrt{\frac{\log |\mathcal{A}_T|}{n}} \longrightarrow \sqrt{\frac{\prod_{k=1}^K M_k}{n}}.$$

In particular, for $|\mathcal{A}_T| = 2^T$ we have $\log |\mathcal{A}_T| = T \log 2$, so the token-level term scales as $\sqrt{T/n}$, while TESS scales as $\sqrt{(\prod_k M_k)/n}$, which is smaller whenever $\prod_k M_k \ll T$.

## B. Detailed Dataset Statistics

*Table 3.* The Electricity dataset represents the half-hourly electricity demand in a state. FNSPID provides daily stock price data integrated with time-aligned financial news. Environment dataset contains daily Air Quality Index (AQI) measurements and Bitcoin denotes the daily Bitcoin price.

| Datasets | Electricity | Bitcoin | FNSPID | Environment |
|---|---|---|---|---|
| Time Horizon | 2019.01-2021.12 | 2019.01-2021.06 | 1999-2023 | 1982-2023 |
| Variates | 19 | 18 | 7 | 4 |
| Timesteps | 52,560 | 858 | 49165 | 11,102 |
| Granularity | 30 minutes | 1 day | 1 day | 1 day |
| Input length | 48 | 7 | 5 | 7 |
| Prediction length | 48 | 7 | 5 | 7 |

We conduct experiments on four real-world multimodal datasets from diverse domains, namely Bitcoin, FNSPID, Electricity, and Environment. Each dataset consists of a target time series accompanied by temporally aligned news text describing relevant external events and contextual information.

The news corpus corresponding to each time series is collected from multiple financial news archives and publicly accessible news repositories, including established sources such as the Reuters News Archive. In addition, we gather supplementary news articles from Google News through third-party aggregation tools, which aggregate and index articles from a wide range of media outlets.

All collected news articles are timestamped and systematically aligned with the corresponding time-series observations based on their publication time, ensuring temporal consistency between the textual and numerical modalities. This alignment enables the model to effectively leverage contemporaneous textual information for time-series forecasting.

