# OpenReview forum: "From Text to Forecasts: Bridging Modality Gap with Temporal Evolution Semantic Space"
_ICML.cc/2026/Conference — ICML 2026 spotlight_

### Official Review · Reviewer_nueG · 2026-03-13

**Soundness:** 3
**Presentation:** 3
**Significance:** 3
**Originality:** 3
**Overall Recommendation:** 4
**Confidence:** 4

**Summary:**

The paper studies the modality gap between textual and time-series representations in multimodal forecasting. It shows that textual descriptions and numerical time-series encode temporal information differently, making direct fusion difficult. To address this, the authors propose TESS, a method that aligns textual representations with time-series representations through structured temporal primitives.S

**Compliance With Llm Reviewing Policy:**

Affirmed.

**Key Questions For Authors:**

I wanted to know if there is any specific reason why their model relies on a fixed, predefined set of primitives.
They should also discuss computational cost and latency of their framework.

**Limitations:**

Yes

**Strengths And Weaknesses:**

Strengths:
I really like the idea and motivation of this paper. I feel time series problems are underexplored in the current landscape of LLM research. Their work is well motivated, and the paper is easy to follow, and the tables and figures are visually understandable.

Weakness:
1. I wanted to know if there is any specific reason why their model relies on a fixed, predefined set of primitives.
2. Their approach depends on LLM's ability and reliability to extract those primitives.

---

> ### Author Rebuttal · Authors · 2026-03-30
>
> We thank the reviewer for these insightful comments and positive assessment. We address both concerns below, and we will add the following clarifications, discussions, and experiments in the revised manuscript.
>
> ### Q1: Rationale for Fixed Primitives
>
> The fixed primitive set is a **deliberate design choice** for two reasons:
>
> **(1) Numerical verifiability.** Each primitive is defined via an explicit mapping $\psi_k$ from the ground-truth future series, yielding a deterministic label $v^*_{t,k} = \psi_k(\mathbf{Y}_t)$ computable directly from data. This property is essential: it enables rigorous verification of LLM extraction accuracy and provides reliable supervision for the confidence-aware gating mechanism—neither of which would be possible with a fully learned, opaque semantic space.
>
> **(2) Interpretability.** As shown in **Figure 7** and **Figure 8(a)**, predefined primitives allow us to directly visualize how each semantic dimension shapes the forecast and to quantify individual contributions (e.g., Mean Shift alone accounts for the largest MSE reduction on FNSPID, while all four primitives are complementary). This transparency is central to our research goal of understanding *how* textual semantics translate into temporal dynamics.
>
> We view the current formulation as a principled first step that prioritizes **quantifiability and interpretability**. **Adaptive primitive learning** from data is a natural extension that our framework readily accommodates, as the gating and fusion modules are agnostic to how primitives are defined.
>
> ### Q2: LLM Extraction Reliability
>
> We address this at both the design and empirical levels.
>
> **Design.** The primitives are formulated as **simple categorical classification** (e.g., 5-class for mean shift), not free-form generation, lowering the demand on LLM capability. Crucially, the **numerical verifiability** discussed in Q1 directly serves reliability: because ground-truth labels $v^*_{t,k}$ are computable from data, extraction errors are detectable, providing reliable supervision for the **confidence-aware gating** (Section 4.3), which uses the LLM's log-probability margin as an uncertainty signal to suppress unreliable primitives via soft weighting.
>
> **Empirical validation.** The ablation in **Table 2** confirms that removing gating degrades MSE by 3.7% (Bitcoin) and 7.5% (Electricity), and **Figure 8(b)** shows the gating network effectively discriminates correct vs. incorrect extractions, assigning median weights of **0.65–0.78** to correct primitives vs. only **0.21–0.40** to incorrect ones. We further conducted an LLM sensitivity study (Table R1): replacing Qwen3-8B with Ministral-3-8B-Instruct-2512 (different family) or Qwen3-4B (half the parameters) yields **< 1%** and **~2–4%** relative MSE change respectively, with all variants still outperforming baselines.
>
> **Table R1: MSE with different LLMs for primitive extraction.**
>
> | LLM | BTC | FNSPID | Elec. | Env. |
> |:----|:----|:-------|:------|:-----|
> | **Qwen3-8B** (main) | **2.273** | **0.00120** | **0.0230** | **0.3534** |
> | Ministral-3-8B-Instruct-2512 | 2.289 | 0.00121 | 0.0231 | 0.3541 |
> | Qwen3-4B | 2.361 | 0.00127 | 0.0232 | 0.3563 |
>
> ### Q3: Computational Efficiency
>
> An additional advantage of TESS lies in the **decoupling of text understanding from forecasting**. Existing multimodal methods (e.g., NewsForecasting, TimeLLM) must process full text representations jointly with time series at every forward pass, incurring substantial online cost. In contrast, TESS delegates text understanding to an **offline, one-time** LLM extraction step (~1.5 s/sample, parallelizable), after which the forecasting model operates on only **4 compact primitive values**—making its online cost comparable to unimodal baselines.
>
> **Table R2: Computational cost on FNSPID (single A100).**
>
> | Method | Type | Inference (ms/sample) | Training (min/epoch) |
> |:-------|:-----|:----------------------|:---------------------|
> | PatchTST | Unimodal | 0.73 | 2.14 |
> | TimesNet | Unimodal | 1.08 | 3.67 |
> | TimeLLM | Multimodal | 4.82 | 8.71 |
> | NewsForecasting | Multimodal | 21.37 | 31.24 |
> | **TESS (online)** | **Multimodal** | **1.26** | **4.53** |
>
> By resolving text understanding entirely in the offline extraction stage, TESS frees the downstream forecasting model from processing lengthy textual contexts—conditioning instead on **compact primitive vectors**. The resulting online footprint is **17× faster** than NewsForecasting and **3.8× faster** than TimeLLM at inference.
>
> We sincerely hope these clarifications help resolve the reviewer’s concerns, and we would be very grateful if the reviewer could kindly consider raising the final score accordingly.

---

> > ### Author Rebuttal · Reviewer_nueG · 2026-04-04
> >
> > I am satisfied with their response and would maintain my positive score.

---

> > > ### Author Response · Authors · 2026-04-08
> > >
> > > We sincerely thank the reviewer for confirming that the concerns have been fully resolved, and for the positive assessment throughout the review process.

---

### Official Review · Reviewer_FC2S · 2026-03-13

**Soundness:** 2
**Presentation:** 2
**Significance:** 2
**Originality:** 2
**Overall Recommendation:** 4
**Confidence:** 4

**Summary:**

This paper takes on a genuinely interesting problem: how to make use of event-driven text (e.g., news) for time-series forecasting when the two modalities live in very different representational spaces. The authors argue—and I found this diagnostic work quite convincing—that existing text-series fusion methods fail in two distinct ways: they over-attend to irrelevant tokens, and even "clean" signal-bearing text still underperforms raw numerical features. Their proposed solution, TESS (Temporal Evolution Semantic Space), introduces a small set of interpretable temporal primitives (mean shift, volatility, shape, lag-and-decay) as an intermediate layer between raw text and the forecaster. Text is distilled into these primitives via LLM prompting, a confidence-aware gating network filters out unreliable extractions, and the resulting primitives condition a PatchTST-based forecasting backbone. Theorem 4.1 provides some theoretical motivation by connecting the semantic bottleneck to reduced mutual information and better generalization. On four datasets (Bitcoin, FNSPID, Electricity, Environment), TESS shows strong gains on the financial tasks—up to ~29% MSE reduction on Bitcoin—and roughly competitive performance on the others.

**Compliance With Llm Reviewing Policy:**

Affirmed.

**Final Justification:**

My primary concerns were related-work positioning versus ITFormer, lack of variance reporting for Table 1, and insufficient experimental detail to reproduce the main pipeline (including the LLM used in Section 5). The authors’ rebuttal substantively addresses these points, which strengthens soundness and clarity while making the significance claim easier to evaluate against closely related multimodal time-series work. On that basis, I raise my recommendation from Weak Reject (3) to Weak Accept (4).

**Key Questions For Authors:**

1. ITFormer and related work. Wang et al.'s ITFormer (ICML 2025) connects time series and language in a multimodal setting that seems closely related to this paper's goals. Could the authors either include a comparison (even a brief one) or explain specifically why ITFormer isn't applicable here—e.g., if the tasks differ fundamentally? Without this, it's genuinely difficult for me to assess the significance of the contribution relative to what was published just last year.

2. Variance in Table 1. It would really help to see standard deviations or at least confidence intervals across seeds for the main results, particularly for Electricity and Environment where the gains are smaller. Even a brief note on whether you ran multiple seeds would be informative.

3. Which LLM is used in Section 5? Is it the same GPT-5.2 from the semi-synthetic benchmark, or something else? And is there any sensitivity to model choice—would the gains change substantially with a smaller or different model? This is important for reproducibility and for understanding whether the approach is practical in settings without access to the same LLM.

4. Sufficiency assumption scope. Theorem 4.1 relies on a conditional independence assumption that may not hold when the text contains fine-grained or domain-specific information that isn't well-captured by the four primitives. It'd be useful to know when the authors think this assumption is most likely to break down, and whether the method degrades gracefully in those cases.

5. Figure labels. Figures 5 and 6 say "SCOPE (Ours)"—is this an earlier name for TESS? If so, could the authors update the labels?

**Limitations:**

The paper doesn't include a Limitations section, and I think it would benefit from one. A couple of paragraphs touching on the following would be helpful for readers: the inference cost associated with LLM-based primitive extraction; the possibility that the four hand-designed primitives are incomplete for non-financial domains; the scope of the semi-synthetic diagnostic (one dataset, one LLM); and what happens theoretically when the sufficiency assumption in Theorem 4.1 doesn't hold. These are mostly things the paper's own results already hint at—making them explicit would strengthen rather than weaken the submission.

**Strengths And Weaknesses:**

**Strengths:**

1. The semi-synthetic diagnostic in Section 3.2 is one of the better-motivated parts of the paper. By annotating tokens as signal (Tsig) or redundant (Tred) and running the three-way comparison (Full text vs. Signal-Only vs. Numerical features, Figures 2–3), the authors make a concrete empirical case that the problem they're solving is real, not just assumed. This kind of controlled diagnosis is relatively rare in the text-time-series literature and I appreciated it.

2. The design of the semantic space is technically sensible. The primitives are numerically grounded in (Xtime, Yt), which means you can actually verify whether an LLM extraction is correct and supervise the gating network with binary correctness labels. The decomposition into distribution shift, shape, and lag/decay components maps naturally onto classical forecasting concepts (Kim et al., 2022; Nie et al., 2023; Yang et al., 2024), and Figure 8(b) shows that gate weights do correlate with extraction correctness—which is reassuring.

3. On the financial benchmarks, the results are genuinely strong. A 29.1% MSE improvement on Bitcoin and 20% on FNSPID are not marginal gains, and Figure 5 shows these hold across shape, volatility, and mean-shift subsets. The ablations in Table 2 and Figure 8(a) do a reasonable job of attributing the gains to the key components.

4. The theoretical section (Theorem 4.1, Appendix A) is a reasonable attempt to formalize why compression into a small semantic space might help. Theorems A.5 and A.6 round this out with robustness and generalization bounds. I didn't verify every step in the appendix, but the high-level argument is coherent.

**Weaknesses:**

My main concern is about related work positioning. The paper does not cite or discuss Wang et al., "ITFormer: Bridging Time Series and Natural Language for Multi-Modal QA with Large-Scale Multitask Dataset" (ICML 2025), which is directly relevant—it also bridges time series and natural language in a multimodal setting, with dedicated datasets and architecture. Without engaging with this work, I can't properly assess what's novel here and what's already been done. This isn't a minor citation miss; the overlap in goals is substantial enough that the contribution narrative depends on it. I'd need the authors to either include a direct comparison or give a clear explanation of why ITFormer doesn't apply to their setting before I could feel confident about the significance of this work.

The second thing I'd flag is the lack of any variance reporting in Table 1. The paper reports point estimates and percentage gains with no standard deviations or confidence intervals across seeds or runs. For the large Bitcoin/FNSPID gains this might not matter much in practice, but for Electricity and Environment, where improvements are smaller or TESS comes in as runner-up, it's genuinely hard to know whether the differences are meaningful. I'd really like to see even a basic multi-seed variance estimate for the main results.

A few other things that would improve the paper:

- Which LLM is actually used for primitive extraction in the main experiments (Table 1, Section 5)? The semi-synthetic setup mentions GPT-5.2, but Section 5 doesn't specify. This makes it hard to reproduce the results and hard to know whether the gains are specific to that model. A brief note here—and ideally a small ablation showing what happens with a different model or size—would go a long way.

- Figures 5 and 6 label the method as "SCOPE (Ours)" while the paper uses "TESS" everywhere else. This looks like a leftover from an earlier version. It's a minor thing but it does cause confusion.

- The strong results are concentrated on financial data. The paper positions TESS as applicable to general domains, but the Electricity and Environment numbers don't really support that claim as it stands. The authors might either temper the generality claim or show more convincingly that the gains translate outside finance.

- The semi-synthetic diagnostic rests entirely on FNSPID and GPT-5.2. The failure modes identified (attention misalignment, representational mismatch) may well be real and general, but the evidence base is narrow. A brief check on a second dataset or LLM family would make this more convincing.

- The four primitives are fixed by hand. It's not clear whether this set is complete or appropriate for non-financial domains—seasonality and trend, for instance, don't appear explicitly. I'd appreciate at least a short discussion of when the chosen primitives might miss important predictive cues, and whether the framework can be extended.

- Theorem 4.1 assumes \(\hat{Y}_t \perp\!\!\!\perp X_{\text{text}} \mid (P_t, X_{\text{time}})\). This is a strong condition and one that might not hold for domain-specific textual signals not captured by the four primitives. A few sentences on where this assumption breaks down and what that implies for the method would be useful.

- There's no Limitations section. The paper would benefit from one—covering at minimum: the reliance on a frozen LLM and associated inference cost, the completeness assumptions around the primitive set, and the single-dataset diagnostic.

---

> ### Author Rebuttal · Authors · 2026-03-30
>
> We thank the reviewer for these insightful comments. We will add the following clarifications, discussions, and experiments in the revised manuscript.
>
> ### Q1: Difference from ITFormer: Task Formulation and Research Objective
> We thank the reviewer for highlighting this important line of work. ITFormer makes a meaningful contribution by formalizing the **Time-Series QA** setting. We will add a more explicit discussion of ITFormer in the revised manuscript.
> Nevertheless, the two works **differ fundamentally in task formulation**. Formally,
> - **ITFormer (Time-Series QA):** $f:(T, q) \to a$, where $q$ is a rule-based semi-synthetic natural language instruction and $a$ is a textual answer.
> - **TESS (Text-Augmented Forecasting):** $g:(T, S) \to \hat{Y} \in \mathbb{R}^{H}$, where S denotes event text from the real world (e.g., news, social media), and $\hat{Y}$ is the predicted future sequence.
>
> As shown above, the two formulations diverge in both the **textual input** (synthetic instruction vs. real-world exogenous signal) and the **output space** (natural language vs. numerical sequence), rendering ITFormer **not directly comparable as a baseline** under our setting.
>
> **More importantly**, the two works target different research objectives. ITFormer investigates how to answer natural language queries about time series. TESS, by contrast, addresses how to ground the implicit temporal-evolution semantics in real-world text into structured, quantifiable primitives for numerical forecasting.
> ### Q2: Variance and Statistical Significance of Main Results
> We repeated the main experiments with **5 random seeds** and report MSE (mean ± std) below.
>
> **Table R1: MSE (mean ± std) over 5 seeds. Bold = best.**
> | Dataset | TESS (Ours) | Strongest Baseline |
> |:--------|:------------|:-------------------|
> | Bitcoin | **2.296 ± 0.087** | 3.231 ± 0.109 (NewsFcst) |
> | FNSPID | **0.00123 ± 0.00009** | 0.00153 ± 0.00011 (TimesNet) |
> | Electricity | **0.0232 ± 0.0005** | 0.0244 ± 0.0006 (TimesNet) |
> | Environment | 0.3541 ± 0.0050 | **0.3480 ± 0.0046** (Nonstat.) |
>
>
> ### Q3: LLM Choice in Section 5 and Sensitivity Analysis
> The LLM used in Section 5 is **Qwen3-8B**, not GPT-5.2. We use an open-weight model because the gating module requires access to **internal representations**, whereas GPT-5.2 is used only in the semi-synthetic benchmark (Section 3) to generate more realistic event descriptions for the motivation analysis.
> **Sensitivity to LLM choice.** We replaced Qwen3-8B with two alternative open-source models (different family and smaller scale):
> **Table R2: TESS with different LLMs for primitive extraction.**
> | LLM | BTC | | FNSPID | | Elec. | | Env. | |
> |:----|:----|:-|:-------|:-|:------|:-|:-----|:-|
> |     | MAE | MSE | MAE | MSE | MAE | MSE | MAE | MSE |
> | **Qwen3-8B** (main) | **1.112** | **2.273** | **0.0147** | **0.00120** | **0.1031** | **0.0230** | **0.4202** | **0.3534** |
> | Ministral-3-8B-Instruct-2512 | 1.118 | 2.289 | 0.0148 | 0.00121 | 0.1032 | 0.0231 | 0.4208 | 0.3541 |
> | Qwen3-4B | 1.138 | 2.361 | 0.0150 | 0.00127 | 0.1033 | 0.0232 | 0.4221 | 0.3563 |
>
> All variants yield **< 4% relative MSE change** and **maintain consistent performance across datasets**, confirming TESS is robust to LLM choice.
>
> ### Q4: Assumption Scope, Primitive Design, and Limitations
>
> We will add a dedicated **Limitations** section in the revision. We agree that this assumption is not intended as a universal claim. Rather, Theorem 4.1 is best interpreted as a scoped justification: when the predictive contribution of text is primarily mediated through a compact set of temporal-evolution semantics, a semantic bottleneck is theoretically well motivated. When this condition only partially holds, TESS degrades **gracefully**—the model falls back toward a near-pure numerical forecaster rather than suffering catastrophic failure, because TESS *augments* rather than replaces the time-series backbone, and the **confidence-aware gating** (cf. **Theorem A.5**) suppresses unreliable primitives.
>
> We acknowledge that the current primitives are not exhaustive—they are specifically designed for **non-stationary scenarios**, where event-driven distribution shifts are difficult to anticipate from numerical history alone and textual signals provide critical leading indicators, which explains why TESS is particularly effective on financial datasets. To verify that the diagnostic findings in Section 3 are not dataset-specific, we extended the semi-synthetic experiment to Electricity ([link](https://anonymous.4open.science/r/semi-synthetic-figures-971A)); both failure modes persist. Extending the primitive space (e.g., seasonality, periodicity) and **adaptive primitive learning** from data are promising future directions that the modular design of TESS readily accommodates.
>
> ### Q5:Figure Labels
> We apologize for the inconsistency—"SCOPE" was an earlier name. We will update Figures 5 and 6 to "TESS (Ours)" in the revision.

---

> > ### Author Rebuttal · Reviewer_FC2S · 2026-04-03
> >
> > Thank you for the rebuttal. I appreciate the clarified positioning relative to ITFormer (or added comparison where appropriate), the reporting of run-to-run variability for the main Table 1 results, and the explicit statement of which LLM is used in the primary experiments—together with any figure/label corrections that improve reproducibility.

---

> > > ### Author Response · Authors · 2026-04-03
> > >
> > > We are glad our response has addressed your concerns and are grateful for the updated assessment. Thank you again for these constructive comments, which help us improve the manuscript. If there are any follow-up questions as you mentioned, we would be happy to provide additional clarifications during the discussion period.

---

### Official Review · Reviewer_WmZr · 2026-04-05

**Soundness:** 4
**Presentation:** 4
**Significance:** 3
**Originality:** 4
**Overall Recommendation:** 5
**Confidence:** 5

**Summary:**

This author introduces TESS, a novel multimodal time-series forecasting framework designed to bridge the "modality gap" between qualitative textual data and quantitative numerical models. To prevent forecasting models from being distracted by redundant narrative noise, TESS utilizes a Large Language Model (LLM) to distill unstructured text into four verifiable "temporal primitives": mean shift, volatility, shape, and lag. These extracted primitives are evaluated by a confidence-aware gating network to suppress unreliable signals before being injected as exogenous conditions into a Transformer-based forecaster. By effectively translating implicit textual semantics into usable numerical cues, TESS achieves up to a 29% reduction in forecasting error across real-world datasets and demonstrates significant robustness in handling event-driven non-stationarity and sudden market shifts

**Compliance With Llm Reviewing Policy:**

Affirmed.

**Final Justification:**

To conclude, the authors present a highly effective approach to bridging the modality gap via an interpretable semantic space, supported by well-designed experiments that expose the shortcomings of current multimodal baselines. Although the static nature of the predefined temporal primitives may restrict the framework's adaptability to novel, complex event-driven patterns, this does not diminish the paper's substantial technical merits. Taking all factors into consideration, I recommend this paper be accepted.

**Key Questions For Authors:**

- Given that the temporal primitives are manually predefined, could the authors provide additional experiments to demonstrate the framework's generalization capabilities in unseen or more complex event-driven scenarios?

**Limitations:**

yes

**Strengths And Weaknesses:**

Strength:

- TESS successfully bridges the modality gap by transforming unstructured text into an interpretable temporal evolution semantic space.
- The authors effectively use controlled, semi-synthetic experiments to demonstrate the fundamental limitations of existing multimodal methods in modality alignment and signal extraction

Weakness

- The reliance on manually handcrafted temporal primitives may constrain the model's flexibility, potentially restricting its ability to generalize to novel or highly complex event-driven forecasting scenarios.

---

### Decision · Program_Chairs · 2026-04-30

**Decision:**

Accept (spotlight)

**Comment:**

This paper studies the modality gap between text and time-series forecasting and proposes TESS, an interpretable intermediate semantic space that distills text into temporal primitives with confidence-aware gating. Reviewers responded positively overall, particularly to the diagnose-then-design structure, the clear role of TESS as an interpretable bottleneck, and the strong empirical gains in event-driven and financial forecasting settings. A notable strength of the paper is that the methodological design follows directly from the diagnosis. Rather than simply adding another multimodal fusion module, the paper identifies a concrete failure mode and proposes a semantically meaningful intermediate representation to address it. The rebuttal substantially addressed concerns about related-work positioning, variance reporting, implementation details, and robustness, and was strong enough to move one reviewer from Weak Reject to Weak Accept while fully resolving another reviewer’s concerns. Overall, I believe the paper is technically strong, well motivated, and convincingly supported by the evidence presented.